# Distinct Expression Patterns of Genes Coding for Biological Response Modifiers Involved in Inflammatory Responses and Development of Fibrosis in Chronic Hepatitis C: Upregulation of SMAD-6 and MMP-8 and Downregulation of CAV-1, CTGF, CEBPB, PLG, TIMP-3, MMP-1, ITGA-1, ITGA-2 and LOX

**DOI:** 10.3390/medicina58121734

**Published:** 2022-11-27

**Authors:** Leona Radmanić, Petra Korać, Lana Gorenec, Petra Šimičić, Kristian Bodulić, Adriana Vince, Snježana Židovec Lepej

**Affiliations:** 1Department of Immunological and Molecular Diagnostics, University Hospital for Infectious Diseases, “Dr. Fran Mihaljević”, HR-10000 Zagreb, Croatia; 2Division of Molecular Biology, Department of Biology, Faculty of Science, University of Zagreb, HR-10000 Zagreb, Croatia; 3Research Department, University Hospital for Infectious Diseases, “Dr. Fran Mihaljević”, HR-10000 Zagreb, Croatia; 4Department of Viral Hepatitis, University Hospital for Infectious Diseases, “Dr. Fran Mihaljević”, HR-10000 Zagreb, Croatia; 5School of Medicine, University of Zagreb, HR-10000 Zagreb, Croatia

**Keywords:** hepatitis C infection, fibrosis, SMAD-6, CAV-1, gene expression

## Abstract

*Background and Objectives*: The aim of this study was to analyze the expression of genes on transcriptomic levels involved in inflammatory immune responses and the development of fibrosis in patients with chronic hepatitis C. *Materials and Methods*: Expression patterns of 84 selected genes were analyzed with real-time quantitative RT PCR arrays in the peripheral blood of treatment-naive patients with chronic hepatitis C and healthy controls. The panel included pro- and anti-fibrotic genes, genes coding for extracellular matrix (EMC) structural constituents and remodeling enzymes, cell adhesion molecules, inflammatory cytokines, chemokines and growth factors, signal transduction members of the transforming growth factor- beta (TGF-ß) superfamily, transcription factors, and genes involved in epithelial to mesenchymal transition. *Results*: The expression of SMAD-6 coding for a signal transduction TGF-beta superfamily member as well as MMP-8 coding for an ECM protein were significantly increased in CHC patients compared with controls. *Conclusions*: Chronic hepatitis C was also characterized by a significant downregulation of a set of genes including CAV-1, CTGF, TIMP-3, MMP-1, ITGA-1, LOX, ITGA-2, PLG and CEBPB encoding various biological response modifiers and transcription factors. Our results suggest that chronic hepatitis C is associated with distinct patterns of gene expression modulation in pathways associated with the regulation of immune responses and development of fibrosis.

## 1. Introduction

Hepatitis C virus (HCV) belongs to the family Flaviviridae, genus Hepacivirus and is an enveloped, positive-sense single-stranded RNA virus [1]. It is most commonly transmitted by direct percutaneous exposure to blood, including drug injection, unsafe health care-associated practices and blood transfusion, while the estimated risk of sexual and vertical transmission remains low [2]. According to the Global Hepatitis Report by the World Health Organization (WHO), an estimated 71.1 million people were chronically infected with HCV in 2015, with a global incidence of HCV infection of 23.7 cases per 100,000 [3]. The clinical course of acute HCV infection is often asymptomatic and remains undiagnosed in the majority of patients. Chronic hepatitis C (CHC) is associated with hepatic inflammation that leads to the development of liver fibrosis and cirrhosis over a period of 10 to 20 years in 10–20% of patients and can subsequently progress to hepatocellular carcinoma (HCC) in approximately 1–4% of patients. Chronic HCV infection is characterized by the direct viral protein-mediated modulation of signaling and metabolic pathways as well as by the induction of antiviral immune responses that subsequently lead to chronic inflammation and the promotion of liver fibrogenesis [2]. Hepatic fibrosis is a complex pathological process that develops as the wound heals in continuing chronic liver injury [4,5,6]. It is mediated by the gathering of extracellular matrix (ECM) and an increased expression of proteins such as collagens, elastin, laminin and fibronectin due to abnormal hyperplasia of connective tissue in the liver. The main initiating event in the development of fibrosis is the activation of hepatic stellate cells (HSCs) that are the primary source of ECM. Activation of ECM is mediated by a variety of biological response modifiers synthesized by hepatocytes and Kuppfer’s cells in response to cell injury including reactive oxygen species, lipid peroxides, inflammatory cytokines and growth factors [4,5,6]. The most important biological response modifiers that are overexpressed during HSC activation and play an important role in the development of fibrosis are platelet-derived growth factors (PDGF)-AA, -AB and -BB and transforming growth factor-β1 (TGF-β1) [4,5,6]. In addition, HSC activation is mediated by complex interactions of immune cells including Th17, Th22, and Th9, mucosa-associated invariant T cells (MAIT), innate lymphoid cells (ILCs) and γδ T cells and their cytokines, particularly IL-17, IL-22 and IL-33. Activated HSCs subsequently differentiate into myofibroblasts that synthesize ECM components, including collagens I, III and α-smooth-muscle actin (α-SMA) [4,5,6]. Highly-effective direct-acting antiviral drugs (inhibitors of HCV polymerase, protease and NS5A) facilitate almost universal viral eradication in persons with CHC, including those with advanced fibrosis, cirrhosis and comorbidities [7]. Despite the evidence of the reversibility of fibrosis following HCV eradication, there is still uncertainty around the treatment’s clinical impact, its predictors, and whether future therapeutic strategies may enhance the probability of fibrosis regression after viral eradication [7,8]. Extensive analysis of molecular patterns that are important for the development and reversal of fibrosis in CHC, both on the local and systemic levels, is expected to play an important role in understanding the physiological background of these events. The aim of this study was to compare the expression of 84 genes involved in the inflammatory immune responses and development of fibrosis on the transcriptomic level, including pro- and anti-fibrotic genes, genes coding for EMC structural constituents and remodeling enzymes, inflammatory cytokines, cell adhesion molecules, growth factors and chemokines, transcription factors, signal transduction members of the TGF-ß superfamily and genes involved in epithelial to mesenchymal transition (EMT). The expression of these genes was compared in peripheral blood samples of patients with CHC and healthy controls.

## 2. Materials and Methods

### 2.1. Subjects and Study Design

This prospective study was conducted at the Department of Viral Hepatitis and the Croatian Reference Centre for Diagnostics and Treatment of Viral Hepatitis at the University Hospital for Infectious Diseases in Zagreb, Croatia. The expression of 84 selected genes was analyzed by performing quantitative PCR in 9 treatment-naive patients with CHC and 3 HCV-negative healthy individuals according to https://dataanalysis2.qiagen.com/pcr (accessed on 20 April 2021) by performing a PCR array analysis. Chronic HCV patients (>18 years) enrolled in this study were not suffering from other infectious or non-infectious chronic diseases, including hepatocellular carcinoma as a complication of chronic CHV infection or pregnancy. Similarly, all healthy individuals included in this study were not suffering from any chronic or acute illness. An assessment of hepatic fibrosis in CHC patients was performed by measuring liver stiffness using transient elastography (FibroScan, Echosens, Paris, France) (F1 < 7.0 kPa, F2 7.0–9.5 kPa, F3 > 9.5 kPa and F4 > 11.5 kPa) [9]. Selected demographic, clinical and virological data (age, gender, HCV genotype, viremia, and risk factors for infection) were extracted from the HCV database of the Croatian Reference Centre for Diagnostics and Treatment of Viral Hepatitis. Ethics committee of UHID approved the study and all subjects signed a consent form.

### 2.2. PCR Array Analysis

According to the manufacturer’s instructions, peripheral blood samples from patients and controls were collected in PAXgene RNA tubes (Pre-AnalytiX, Hombrechtikon, Switzerland) and RNA was isolated using the PAXgene RNA Kit (Pre-AnalytiX, Hombrechtikon, Switzerland). The quality and concentration of isolated RNA were tested using a BioPhotometer pPlus spectrophotometer (Eppendorf, Hamburg, Germany) by measuring the absorbance at wavelengths of 230 nm, 260 nm and 280 nm. cDNA was synthesized using the QIAGEN RT2 First Strand Kit (QIAGEN, Hilden, Germany) following the manufacturer’s instructions and added to the RT2 SYBR Green qPCR Master Mix (QIAGEN, Hilden, Germany). The samples were added to the 96-well plate of RT 2 Profiler TM PCR Array Human Fibrosis (QIAGEN, Hilden, Germany). The expression of 84 genes involved in the immune response and the pathogenesis of fibrosis was determined using real-time quantitative RT-PCR. PCR reactions were carried out by using ABI 7500 Fast (Applied Biosystems, Foster City, CA, USA) under the following conditions: 10 min on 95 C, 40 cycles for 15 s on 95 C and 1 min on 60 C. Calculations and analysis of Ct values and melting curves for all PCR reactions were performed using 7500 System SDS Software v1.4.0 (Applied Biosystems, Foster City, CA, USA). The PCR array contained a panel of 96 primer sets. A total of 84 sets were specific for fibrosis genes, 5 for housekeeping genes, 1 for genomic DNA control, 3 for reverse transcription controls and 3 for positive PCR controls. The array facilitated the analysis of the following genes: pro-fibrotic genes (AGT, CCL-11 (Eotaxin) and ACTA-2 (α-SMA)), CCL-2 (MCP-1), CCN-2 (CTGF), CCL-3 (MIP-1A), GREM-1, IL-13, IL-5, IL-13RA-2, IL-4, SNAI-1 (SNAIL); anti-fibrotic genes (IFNG, IL-10, BMP-7, HGF and IL-13RA-2); extracellular matrix (ECM) and cell adhesion molecules, extracellular matrix (ECM) structural constituents (COL-1A2 and COL-3A1); extracellular matrix (ECM) remodeling enzymes (LOX, MMP-1, MMP-2, MMP-3, MMP-13, MMP-14, MMP-8, MMP-9, PLG, PLAT (TPA), PLAU (UPA), SERPINH-1 (HSP47), SERPINA-1, SERPINE-1 (PAI-1), TIMP-4, TIMP-3, TIMP-2 and TIMP-1); cell adhesion molecules (ITGA-3, ITGA-2, ITGA-1, ITGAV, ITGB-3, ITGB-1, ITGB-6, ITGB-8 and ITGB-5), inflammatory cytokines and chemokines (CCL-11 (Eotaxin), CCL-3 (MIP-1A), IL-5, CCL-2 (MCP-1), TNF, CCR-2, IL-13, CXCR-4, IFNG, IL-1A, IL-10, IL13-RA-2, IL-1B, IL-4 and ILK); growth factors (EDN-1, AGT, EGF, CCN-2, HGF, PDGFB, PDGFA, VEGFA); signal transduction TGF-ß superfamily members (CAV-1, BMP-7, GREM-1, DCN, LTBP-1, ENG (EVI-1), INHBE, SMAD-7, SMAD-2 (MADH2), SMAD-3 (MADH3), SMAD-4 (MADH4), SMAD-6, THBS-1 (TSP-1), THBS-2, TGFB-3, TGFB-2, TGFB-1, TGFBR-1 (ALK5), TGFBR-2 and TGIF-1); transcription factors (JUN, CEBPB, MYC, SP-1, STAT-1, NFKB-1 and STAT-6); epithelial to mesenchymal transition (EMT) (COL-1A2, AKT-1, ILK, BMP-7, COL-3A1, ITGAV, ITGB-1, MMP-3, MMP-9, MMP-2, SERPINE-1(PAI-2), SMAD-2 (MADH2), SNAI (SNAIL), TGFB-3, TGFB-1, TGFB-2 and TIMP-1); and other fibrosis genes (FASLG (TNFSF6) and BCL-2,). Data analysis was performed using RT2 Profiler PCR Array Data Analysis v3.5 (QIAGEN, Hilden, Germany). Toidentify and visualize significant changes in gene expression. a volcano plot was created.

### 2.3. Statistical Analysis

Data were analyzed using RT2 Profiler PCR Array Data Analysis (QIAGEN, Hilden, Germany). The Student’s *t*-test was used to determine significant change in expression between HCV patients and controls. Results with a *p* value lower than 0.05 were considered statistically significant. The *p*-values were corrected for multiple testing with the Benjamini–Hochberg method.

## 3. Results

### 3.1. Expression of Genes Coding for Mediators of Inflammatory Responses and Development of Fibrosis in CHC

The levels of expression for selected genes were compared in nine CHC patients (five males and four females median age 47, range 21–53 years, median viremia 1,862,000 IU/mL, range 51,337–11,883,725 IU/mL, F4 *n* = 3, F3 *n* = 1, F2 *n* = 3 and F1 *n* = 2,) and healthy controls (*n* = 3, median age 29, range 28–41 years, three females). Detailed participant characteristics are shown in Table 1. Expression of the SMAD-6 gene was significantly increased and the expression of the LOX, CAV-1, PLG, ITGA-2, MMP-1, and CTGF genes was significantly decreased in CHC patients compared to controls (Figure 1, Table 2).

### 3.2. Expression of Genes Coding for Mediators of Inflammatory Responses and Development of Fibrosis in Patients with Mild Fibrosis (F1/F2)

Gene expression patterns in patients with CHC and mild fibrosis (two patients with F1 and three patients with F2) showed a statistically significant downregulation of CAV-1, CEBPB and ITGA-1 compared with controls (Figure 2, Table 3).

### 3.3. Expression of Genes Coding for Mediators of Inflammatory Responses and Development of Fibrosis in Patients with Advanced Fibrosis (F3/F4)

Gene expression patterns in patients with CHC and advanced fibrosis (one patient with F3 and three patients with F4) showed a statistically significant upregulation of the MMP-8 gene as well as significant downregulation of the CAV-1, CTGF, LOX, PLG and TIMP-3 genes compared with controls (Figure 3, Table 4).

## 4. Discussion

The results of this study reveal important differences in the expression patterns of 84 genes coding for biological response modifiers that are important for the inflammatory immune responses and immunopathogenesis of fibrosis in patients with CHC compared with healthy controls on the transcriptomic level, since most previous research on this topic used immunohistochemical methods. Increased expression of SMAD-6 was a common molecular signature in patients with CHC compared with healthy controls. SMADs are a family of signal transducers and activators of transcription that mediate transforming growth factor beta (TGF-β) signaling pathways [10]. The importance of SMAD-6 in HCV infection, particularly during the viral attachment stage, was demonstrated by Zhang et al., (2017) who demonstrated that HCV infection of hepatoma cell lines and human primary hepatocytes upregulated the expression of SMAD-6 via the activation of NF-kappaB [11]. The upregulation of SMAD-6 was associated with an increased expression of cell surface heparan sulfate proteoglycans (HSPGs) that play an important role in the complex steps leading to HCV entry into hepatocytes, as well as with the uptake of cholesterol and lipoproteins in target cells. Zhang et al., (2017) also showed an increased expression of SMAD-6 and SMAD-7 genes in liver biopsies from HCV-infected patients [11,12]. Given the pleiotropic nature of TGF-β pathways in inflammation, immune regulation, the development of fibrosis and oncogenesis, the biological and possible clinical importance of increased SMAD-6 expression in HCV infection requires further investigation [13,14]. Our results also showed a pronounced downregulation of the CAV-1 gene in HCV-infected patients compared to controls, particularly those with advanced fibrosis. Caveolin-1 is a major component of caveolae and plays an important role in lipid trafficking and metabolism [15,16]. CAV-1 is also an important regulator of several cell signaling pathways and plays an important role in malignant cell transformation, tumor growth, angiogenesis and metastatic processes [16]. An increased expression of CAV-1 on malignant epithelial cells and the loss of stromal CAV-1 expression on cancer-associated fibroblasts are important indicators of disease progression and unfavorable clinical outcomes in solid tumors [16]. The majority of early evidence on the possible involvement of CAV-1 in the pathogenesis of CHC is based on experiments showing an increased expression of this molecule on caveolae-like structures within liver sinusoidal endothelial cells (LSEC) undergoing capillarization that usually precede the development of liver fibrosis [17]. In addition, the increased expression of CAV-1 was found in cirrhotic livers of patients infected with HCV; however, this finding is probably associated with higher cholesterol levels in cirrhosis [18]. Different patterns of CAV-1 expression in hepatocellular carcinoma suggest that this molecule plays an important role in tumor progression too [19]. More recently, Baiocchini et al., (2019) described different patterns of CAV-1 expression on LSEC and hepatocytes in the liver of HCV-infected persons vs. healthy controls, as well as a significant association between the nuclear expression of CAV-1 in hepatocytes and the stage of fibrosis [20]. Interestingly, Gao et al., (2021) showed that miR-629-5p promotes osteosarcoma proliferation and migration via the direct inhibition of CAV-1 mRNA expression [21]. In our study, we noticed that the expression of CAV-1 was downregulated when we compared F1/F2 stages of fibrosis with healthy controls and in F3/F4 stages of fibrosis compared with healthy controls. These results suggest that the impaired function of CAV-1 as a tumor suppressor gene may contribute to the exacerbation of liver fibrosis and ultimately the development of HCC; this requires further research. Matrix metalloproteinases (MMPs), a family of zinc-dependent extracellular matrix (ECM) remodeling endopeptidases and tissue inhibitors of MMPs (TIMPs), play an important role in a variety of physiological processes, including the development of fibrosis [22,23]. The results of our study demonstrate increased MMP-8 expression in CHC patients with F3/F4 stages of fibrosis compared to healthy controls. These results are in agreement with the study of Capone et al., (2012) showing increased concentrations of MMP-8, MMP-9, TRAIL and β-NGF in the serum of HCV infected patients compared with controls [24]. MMP-1 encodes the synthesis of matrix metalloproteinase 1 (MMP-1), an enzyme responsible for the degradation of collagen-I. Several studies have shown an association between reduced serum concentrations of MMP-1 and progression of liver fibrosis [25,26]. These findings are in agreement with our results showing the significant downregulation of MMP-1 expression in all stages of fibrosis compared to healthy controls. In addition, reduced expression of TIMP-3 that inhibits the proteolytic activity of MMPs observed in CHC patients with F3/F4 suggests a possible shift towards maintenance of the ECM structure in patients with fibrosis [23]. Connective tissue growth factor (CTGF/CCN2) is a matricellular protein that plays an important role in regulating cellular differentiation, proliferation, adhesion, migration and apoptosis as well as production of the extracellular matrix, wound healing and fibrosis [Huang et al., 2012; Tam et al., 2021]. The expression of CTGF by hepatic stellate cells is considered to be one of the key events in the pathogenesis of liver fibrosis [27,28]. The downregulation of CTGF observed in this study is probably associated with the modulation of TGF-ß signaling pathways through overexpression of SMAD-6, but it would be interesting to perform a transcriptomic analysis in a larger cohort. The results of our study also demonstrate the significant downregulation of several genes (ITGA-1, ITGA-2, LOX, CEBPB and PLG) coding for ECM and cell adhesion molecules in patients with CHC compared to controls. Batyrova et al., (2019) recently showed slightly higher levels of PGL in CHC patients compared with chronic hepatitis B; however, the significance of this finding needs to be further evaluated [29]. The ITGA-2 gene plays a critical role in various developmental processes, including the differentiation, proliferation, and migration of cells [30]. LOX encodes the enzyme lysyl oxidase, which enables the cross-linking of collagen and elastin fibers within the ECM [31]. Li et al., (2018) demonstrated that CEBPB is a transcription factor which possibly plays an important role in inflammatory status after HCV infection [32]. The literature data clearly suggest that the downregulation of ITGA-1, ITGA-2, CEBPB and LOX genes in chronic HCV infection needs to be evaluated further on transcriptomic levels.

Several limitations of this study should be mentioned. The sample size of the healthy control group is notably low, resulting in the low statistical power observed of the performed statistical tests. This number was sufficient to detect a substantial amount of differentially expressed genes. Additionally, the exclusion criteria used in this study resulted in a lower biological variation in gene expression and increased the power of performed comparisons. However, the differential expression of the stated genes should be evaluated in future research due to the possibility of false-positive results. Furthermore, gene expression levels were measured in peripheral blood samples, implying that extrapolation of our results to liver tissue should be done with caution. However, different studies suggest that the immunomodulatory effects of CHC infection are detectable on the systemic level, especially in the case of CHC patients without other underlying conditions [24,33,34]. Finally, there was a significant difference in sex distribution between the patient and healthy control groups, with all healthy controls being female. Considering that we did not find any differentially expressed genes in male and female patients, we believe that this limitation did not significantly impact the conclusions of our study. This is further supported by studies demonstrating a lack of association between sex and the expression of CAV-1 and MMP-1 in patients with CHC or CHC-related complications, such as hepatocellular carcinoma [19,35].

## 5. Conclusions

In conclusion, the results of this study demonstrate distinct expression patterns of genes coding for biological response modifiers involved in inflammatory responses and the pathogenesis of fibrosis in CHC. In peripheral blood samples, chronic HCV infection is characterized by opposing effects in terms of the expression of signal transduction in TGF-beta superfamily members CAV-1 and SMAD-6 and an overexpression of the MMP-8 gene coding for an EMC remodeling enzyme; however, further evaluations are necessary. In addition, the downregulation of a distinct set of genes including CTGF, LOX, MMP-1, PLG, TIMP-3, ITGA-1, ITGA-2 and CEBPB belonging to diverse biological pathways reveals the complexity of the molecular mechanisms involved in the pathogenesis of inflammation and fibrosis in CHC. The results of this study also suggest that, following viral eradication by DAA, novel therapeutic strategies focusing on the reversal of fibrosis and inflammatory responses might require a multi-targeted approach and further investigation on transcriptomic levels [36].

## Figures and Tables

**Figure 1 medicina-58-01734-f001:**
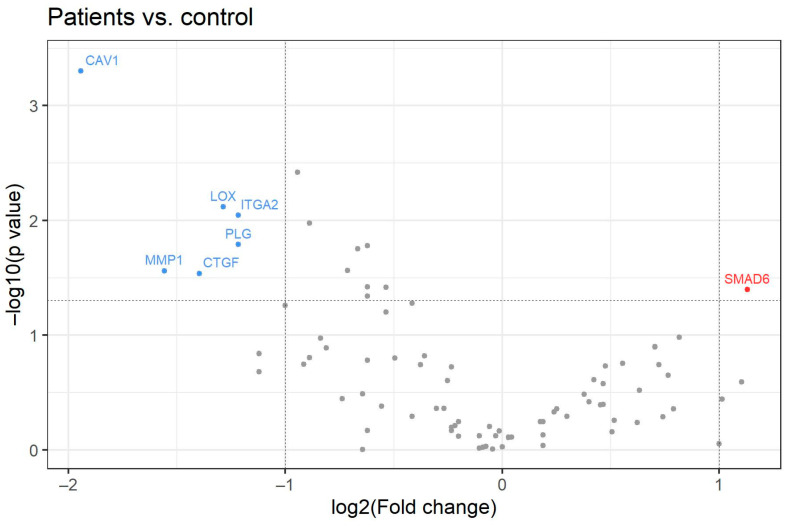
Volcano plot for 84 analyzed genes. The X-axis shows the log2 value of the fold change between values for patients with chronic hepatitis C compared to healthy controls. The Y-axis represents −log10 *p*-value. Red points symbolize overexpressed genes and blue points represent underexpressed genes. The horizontal line shows the threshold where *p* = 0.05.

**Figure 2 medicina-58-01734-f002:**
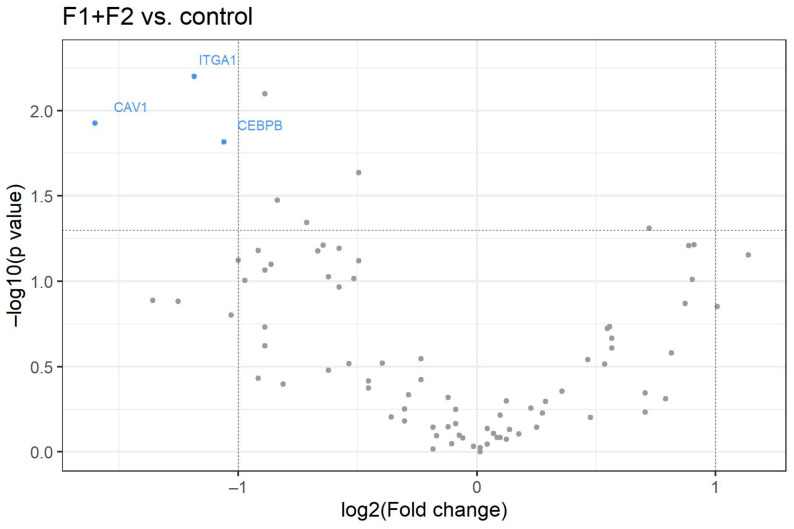
Volcano plot for 84 analyzed genes. The X-axis shows the log2 value of the fold change between patients with F1 and F2 stages of chronic hepatitis C compared to healthy controls without HCV. The Y-axis represents −log10 *p*-value. Red points symbolize overexpressed genes and blue points represent underexpressed genes. Horizontal line shows threshold where *p* = 0.05.

**Figure 3 medicina-58-01734-f003:**
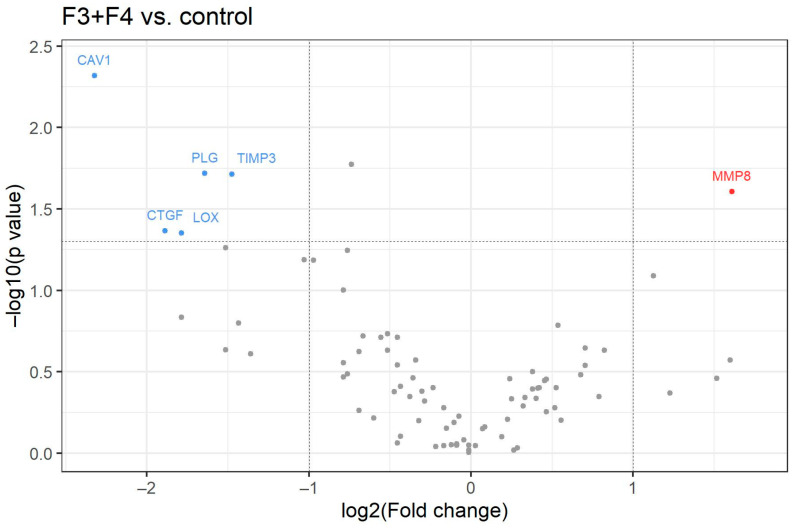
Volcano plot for 84 analyzed genes. The X-axis shows the log2 value of the fold change between patients with F3 and F4 stages of chronic hepatitis C compared to the healthy controls without HCV. The Y-axis represents −log10 *p*-value. Red points symbolize overexpressed genes and blue points represent underexpressed genes. Horizontal line shows threshold where *p* = 0.05.

**Table 1 medicina-58-01734-t001:** Demographic, clinical and virology data of the study population.

Participant	Age (Years)	Gender	HCV Genotype (Subtype)	HCV Viremia (IU/mL)	Stage of Fibrosis	Risk Factors
P1	53	female	3a	7,381,324	F4	operation, tattoo
P2	53	male	3a	252,281	F4	IDU
P3	47	male	1b	2,400,000	F2	transfusion
P4	44	female	1a	1,862,000	F2	transfusion
P5	41	male	3a	11,883,725	F3	IDU
P6	51	female	3a	295,367	F1	transfusion
P7	38	male	3a	3,257,675	F2	IDU
P8	47	male	3a	236,648	F4	operation
P9	21	female	3a	51,337	F1	IDU
HC1	41	female	/	/	/	/
HC2	29	female	/	/	/	/
HC3	28	female	/	/	/	/

HCV = hepatitis C virus, P = patient, HC = healthy control, IDU-intravenous drug use.

**Table 2 medicina-58-01734-t002:** Gene expression patterns in patients with chronic hepatitis C and controls. P-values are corrected for multiple testing with Benjamini–Hochberg method.

Gene	Effect on Gene Expression	Fold Regulation	*p*-Value	Adjusted *p*-Value
CAV-1	-	−3.81	0.001	0.04
PLG	-	−2.34	0.02	0.19
CTGF	-	−2.65	0.03	0.22
MMP-1	-	−2.92	0.03	0.22
SMAD-6	+	2.19	0.04	0.24
LOX	-	−2.44	0.008	0.18
ITGA-2	-	−2.34	0.009	0.18
ITGB-8	-	−2.16	0.16	0.47
IL-5	-	−2.18	0.21	0.50
MMP-8	+	2.15	0.26	0.55
IL-13	+	2.02	0.36	0.68

**Table 3 medicina-58-01734-t003:** Gene expression patterns in patients with chronic hepatitis C with mild fibrosis (F1/2) and controls. *p*-values were corrected for multiple testing with Benjamini–Hochberg method.

Gene	Effect on Gene Expression	Fold Regulation	*p*-Value	Adjusted *p*-Value
CAV-1	-	−3.06	0.01	0.32
ITGA-1	-	−2.27	0.01	0.32
CEBPB	-	−2.08	0.02	0.32
SMAD-6	+	2.20	0.07	0.35
TIMP-4	+	2.20	0.07	0.35
ITGA-2	-	−2.01	0.08	0.35
MMP-1	-	−2.58	0.13	0.41
SERPINE-1	-	−2.35	0.13	0.41
GREM-1	+	2.01	0.14	0.41
CTGF	-	−2.05	0.16	0.44

**Table 4 medicina-58-01734-t004:** Gene expression patterns in patients with chronic hepatitis C with mild fibrosis (F1/2) and controls. *p*-values were corrected for multiple testing with Benjamini–Hochberg method.

Gene	Effect on Gene Expression	Fold Regulation	*p*-Value	Adjusted *p*-Value
CAV-1	-	−5.01	0.004	0.40
MMP-8	+	3.05	0.02	0.42
PLG	-	−3.12	0.02	0.41
TIMP-3	-	−2.78	0.02	0.41
CTGF	-	−3.64	0.04	0.50
LOX	-	−3.40	0.04	0.50
ITGA-2	-	−2.82	0.05	0.50
TGFBR-2	-	−2.03	0.07	0.50
SMAD-6	+	2.18	0.08	0.57
MMP-1	-	−3.41	0.15	0.72
SMAD-3	-	−2.67	0.16	0.72
IL-5	-	−2.85	0.23	0.72
ITGB-8	-	−2.57	0.24	0.72
IL-13	+	3.03	0.27	0.72
DCN	+	2.86	0.35	0.72
COL-3A1	+	2.34	0.43	0.72

## Data Availability

The data presented in this study are available on request from the corresponding author. The data are not publicly available due to privacy and ethical restrictions.

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
