# Peer review of "Distinct Expression Patterns of Genes Coding for Biological Response Modifiers Involved in Inflammatory Responses and Development of Fibrosis in Chronic Hepatitis C: Upregulation of SMAD-6 and MMP-8 and Downregulation of CAV-1, CTGF, CEBPB, PLG, TIMP-3, MMP-1, ITGA-1, ITGA-2 and LOX"

_medicina, 2022, doi:10.3390/medicina58121734_

Round 1

Reviewer 1 Report

The authors analysed the expression of genes on transcriptomic levels involved in inflammatory immune responses and the development of fibrosis in patients with chronic hepatitis C.Finally, they concluded that  Chronic hepatitis C is associated with distinct patterns of gene expression modulation in pathways related to the regulation of immune responses and development of fibrosis.

The authors selected nine patients with chronic hepatitis C and compared them to three healthy individuals with negative hepatitis C . The authors have to mention how they calculated the sample size and justify why the number of healthy controls is sufficient.

They should also evolve the inclusion and exclusion criteria of the study.

The authors mentioned that they extracted the participants' demographic, clinical and virology data, and they should present these data briefly in one table.

The authors mentioned that they did a hepatic fibrosis assessment. They should briefly mention the evaluation method and the number of participants in each and justify the suitability of the number in each class.

The authors discussed the data excellently, but they should mention the limitations of their study

Author Response

Reviewer 1

We would like to thank the reviewer for helpful suggestions and the time taken to review our manuscript. Here are our remarks.

The authors selected nine patients with chronic hepatitis C and compared them to three healthy individuals with negative hepatitis C . The authors have to mention how they calculated the sample size and justify why the number of healthy controls is sufficient.

We thank you for the comment regarding the analysed sample size. The number of test subjects chosen for this experiment allowed sufficient power to detect a substantial amount of differentially expressed genes while controlling for biological and technical variation in every experimental group. This is supported by various reviews and RT-PCR protocols which suggest a minimal number of three biological replicates per experimental state. Furthermore, numerous gene expression studies used lower sample sizes and produced statistically significant and biologically relevant results. Additionally, the exclusion criteria for patients in this study ensured a relatively low biological variation in the expression of analysed genes. Chronic HCV patients enrolled in this study were not suffering from other infectious or non-infectious chronic diseases, hepatocellular carcinoma as a complication of chronic CHV infection or pregnancy. Similarly, all healthy individuals included in this study were not suffering from any chronic or acute illness and had similar age ranges compared to the patient group. While the stated reduction in biological variation increased statistical power, we do agree that the inclusion of more samples would allow for more powerful statistical comparisons, which could potentially reveal more differentially expressed genes. Accordingly, we revised the discussion of our manuscript to emphasize this limitation and the potential of future research of genes differentially expressed in our study (lines 289-299).

They should also evolve the inclusion and exclusion criteria of the study.

We thank you for your suggestion, so we involved the inclusion and exclusion criteria of the study (lines 94-97).

The authors mentioned that they extracted the participants' demographic, clinical and virology data, and they should present these data briefly in one table.

We added a table in the manuscript with all the participants' and their demographic, clinical and virology data (Table 1) (line 167).

The authors mentioned that they did a hepatic fibrosis assessment. They should briefly mention the evaluation method and the number of participants in each and justify the suitability of the number in each class.

We mentioned in the manuscript that assessment of hepatic fibrosis in CHC patients was performed by measuring liver stiffness using transient elastography (FibroScan, Echosens, France) (F1<7.0 kPa, F2 7.0–9.5 kPa, F3 > 9.5 kPa and F4 > 11.5 kPa) [9] (lines 97-100). In Table 1, we added the stage of fibrosis of each patient (line 167).

The authors discussed the data excellently, but they should mention the limitations of their study.

We thank you for your suggestion, we added the limitations of our study at the end of discussion (lines 289-299):

Several limitations of this study should be mentioned. The sample size of the healthy control group is notably low, resulting in the low statistical power of performed statistical tests. This number was sufficient to detect a substantial amount of differentially expressed genes. However, the differential expression of the stated genes should be evaluated in future research due to the possibility of false-positive results. Additionally, the exclusion criteria used in this study resulted in lower biological variation of gene expression and increased the power of performed comparisons. Furthermore, gene expression levels were measured in peripheral blood samples, implying that extrapolation of our results to liver tissue should be done with caution. However, different studies suggest that immunomodulatory effects of chronic HCV infection are detectable on the systemic level, especially in the case of chronic HCV patients without other underlying conditions.

Reviewer 2 Report

There is no doubt that research in the field of inflammatory processes connected with the chronic hepatitis C infection is very important. But I’m worried that the study design chosen by authors is not optimal to achieve the goal.

1)      In the descripted study the expression patterns of the genes are detected not in the liver tissues, but in the peripheral blood cells. Sure, there are evidences that immunomodulatory effects of chronic HCV infection can be detected also on the systemic level, but it is necessary to be very careful extrapolating that data to the processes in the liver.  Authors did very good work comparing their results with the literature data. Reading the “Discussion” section I got impression that findings of the Authors are in good concordance with the results of another blood-based studies but sometimes differ from the data obtained studying the liver tissues of even the cell culture models (like in the case of CAV-1 gene). I think that shall be reflected in the discussion and in conclusion. Also, I think the abstract of the manuscript must be updated with the statement that the object of study is the peripheral blood cells to make it clearer for the readers.

2)      The patients’ cohorts used by authors are extremely small. I understand that there can be difficult to find the treatment-naïve chronic HCV patients, but the “healthy” persons group definitely must be larger than just 3. The age and gender distribution in both healthy and chronic HCV patients’ groups have must better accordance to avoid the possibility of bias.

3)      Description of the persons included in the study must be improved. Besides the chronic HCV infection there are number of other factors influencing the systemic inflammation process, like immune status and autoimmune diseases, alcohol and drugs using, specific medications (not necessarily anti-viral), background acute and chronic infections, etc. It is important not only for the chronic HCV patients, but also for the healthy persons group used as control, especially if used cohorts are so small.

In the conclusion, I think that the manuscript has a potential, but needs lot of improvements.

Author Response

Reviewer 2

We would like to thank the reviewer for helpful suggestions and the time taken to review our manuscript. Here are our remarks.

1) In the descripted study the expression patterns of the genes are detected not in the liver tissues, but in the peripheral blood cells. Sure, there are evidences that immunomodulatory effects of chronic HCV infection can be detected also on the systemic level, but it is necessary to be very careful extrapolating that data to the processes in the liver. Authors did very good work comparing their results with the literature data. Reading the “Discussion” section I got impression that findings of the Authors are in good concordance with the results of another blood-based studies but sometimes differ from the data obtained studying the liver tissues of even the cell culture models (like in the case of CAV-1 gene). I think that shall be reflected in the discussion and in conclusion. Also, I think the abstract of the manuscript must be updated with the statement that the object of study is the peripheral blood cells to make it clearer for the readers.

We thank the reviewer for this remark. We emphasized in the abstract that the object of this  study is gene expression in the the peripheral blood cells to make it clearer for the readers. Furthermore, in discussion and conclusion we added that gene expression levels were measured in peripheral blood samples, implying that extrapolation of our results to liver tissue should be done with caution.

2) The patients’ cohorts used by authors are extremely small. I understand that there can be difficult to find the treatment-naïve chronic HCV patients, but the “healthy” persons group definitely must be larger than just 3. The age and gender distribution in both healthy and chronic HCV patients’ groups have must better accordance to avoid the possibility of bias.

We thank you for the comment regarding the analysed sample size. The number of test subjects chosen for this experiment allowed sufficient power to detect a substantial amount of differentially expressed genes while controlling for biological and technical variation in every experimental group. This is supported by various reviews and RT-PCR protocols which suggest a minimal number of three biological replicates per experimental state. Additionally, the exclusion criteria for patients in this study ensured a relatively low biological variation in the expression of analysed genes. Chronic HCV patients enrolled in this study were not suffering from other infectious or non-infectious chronic diseases, hepatocellular carcinoma as a complication of chronic CHV infection or pregnancy. Similarly, all healthy individuals included in this study were not suffering from any chronic or acute illness and had similar age ranges compared to the patient group. While the stated reduction in biological variation increased statistical power, we do agree that the inclusion of more samples would allow for more powerful statistical comparisons, which could potentially reveal more differentially expressed genes. Accordingly, we revised the discussion of our manuscript to emphasize this limitation and the potential of future research of genes differentially expressed in our study (lines 289-299).

3) Description of the persons included in the study must be improved. Besides the chronic HCV infection there are number of other factors influencing the systemic inflammation process, like immune status and autoimmune diseases, alcohol and drugs using, specific medications (not necessarily anti-viral), background acute and chronic infections, etc. It is important not only for the chronic HCV patients, but also for the healthy persons group used as control, especially if used cohorts are so small.

We added a table in the manuscript with all the participants' and theirs demographic, clinical and virology data (Table 1) (line 167). Also, we included the inclusion and exclusion criteria of the study and limitations (lines 94-97).